# SELFCHECKGPT: Zero-Resource Black-Box Hallucination Detection for Generative Large Language Models

**Potsawee Manakul, Adian Liusie, Mark J. F. Gales**
ALTA Institute, Department of Engineering, University of Cambridge
pm574@cam.ac.uk, al826@cam.ac.uk, mjfg@eng.cam.ac.uk

## Abstract

Generative Large Language Models (LLMs) such as GPT-3 are capable of generating highly fluent responses to a wide variety of user prompts. However, LLMs are known to hallucinate facts and make non-factual statements which can undermine trust in their output. Existing fact-checking approaches either require access to the output probability distribution (which may not be available for systems such as ChatGPT) or external databases that are interfaced via separate, often complex, modules. In this work, we propose "SelfCheckGPT", a simple sampling-based approach that can be used to fact-check the responses of black-box models in a zero-resource fashion, i.e. without an external database. SelfCheckGPT leverages the simple idea that if an LLM has knowledge of a given concept, sampled responses are likely to be similar and contain consistent facts. However, for hallucinated facts, stochastically sampled responses are likely to diverge and contradict one another. We investigate this approach by using GPT-3 to generate passages about individuals from the WikiBio dataset, and manually annotate the factuality of the generated passages. We demonstrate that SelfCheckGPT can: i) detect non-factual and factual sentences; and ii) rank passages in terms of factuality. We compare our approach to several baselines and show that our approach has considerably higher AUC-PR scores in sentence-level hallucination detection and higher correlation scores in passage-level factuality assessment compared to grey-box methods.[1]

## 1 Introduction

Large Language Models (LLMs) such as GPT-3 (Brown et al., 2020) and PaLM (Chowdhery et al., 2022) are capable of generating fluent and realistic responses to a variety of user prompts. They have been used in many applications such as automatic

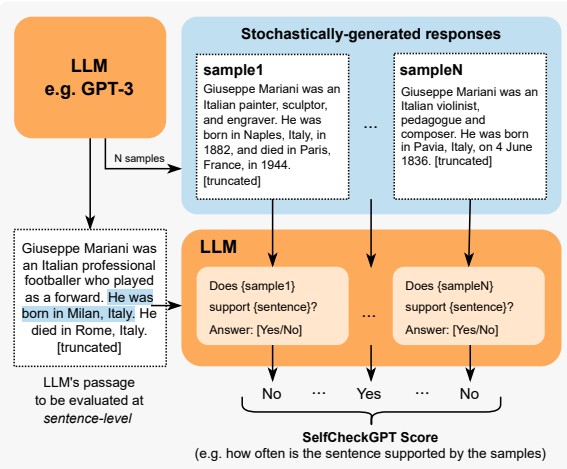

**Figure 1:** SelfCheckGPT with Prompt. Each LLM-generated sentence is compared against stochastically generated responses with no external database. A comparison method can be, for example, through LLM prompting as shown above.

tools to draft reports, virtual assistants and summarization systems. Despite the convincing and realistic nature of LLM-generated texts, a growing concern with LLMs is their tendency to hallucinate facts. It has been widely observed that models can confidently generate fictitious information, and worryingly there are few, if any, existing approaches to suitably identify LLM hallucinations.

A possible approach of hallucination detection is to leverage existing intrinsic uncertainty metrics to determine the parts of the output sequence that the system is least certain of (Yuan et al., 2021; Fu et al., 2023). However, uncertainty metrics such as token probability or entropy require access to token-level probability distributions, information which may not be available to users for example when systems are accessed through limited external APIs. An alternate approach is to leverage fact-verification approaches, where evidence is retrieved from an external database to assess the veracity of a claim (Thorne et al., 2018; Guo et al., 2022). However, facts can only be assessed relative to the knowledge present in the database. Addition-

---

[1]Code and dataset can be found on the project page at https://github.com/potsawee/selfcheckgpt.

ally, hallucinations are observed over a wide range of tasks beyond pure fact verification (Kryscinski et al., 2020; Maynez et al., 2020).

In this paper, we propose SelfCheckGPT, a sampling-based approach that can detect whether responses generated by LLMs are hallucinated or factual. To the best of our knowledge, SelfCheck-GPT is the first work to analyze model hallucination of general LLM responses, and is the first zero-resource hallucination detection solution that can be applied to black-box systems. The motivating idea of SelfCheckGPT is that when an LLM has been trained on a given concept, the sampled responses are likely to be similar and contain consistent facts. However, for hallucinated facts, stochastically sampled responses are likely to diverge and may contradict one another. By sampling multiple responses from an LLM, one can measure information consistency between the different responses and determine if statements are factual or hallucinated. Since SelfCheckGPT only leverages sampled responses, it has the added benefit that it can be used for black-box models, and it requires no external database. Five variants of SelfCheckGPT for measuring informational consistency are considered: BERTScore, question-answering, $n$-gram, NLI, and LLM prompting. Through analysis of annotated articles generated by GPT-3, we show that SelfCheckGPT is a highly effective hallucination detection method that can even outperform grey-box methods, and serves as a strong first baseline for an increasingly important problem of LLMs.

## 2 Background and Related Work

### 2.1 Hallucination of Large Language Models

Hallucination has been studied in text generation tasks, including summarization (Huang et al., 2021) and dialogue generation (Shuster et al., 2021), as well as in a variety of other natural language generation tasks (Ji et al., 2023). Self-consistency decoding has shown to improve chain-of-thought prompting performance on complex reasoning tasks (Wang et al., 2023). Further, Liu et al. (2022) introduce a hallucination detection dataset, however, texts are obtained by perturbing factual texts and thus may not reflect true LLM hallucination.

Recently, Azaria and Mitchell (2023) trained a multi-layer perception classifier where an LLM's hidden representations are used as inputs to predict the truthfulness of a sentence. However, this approach is a white-box approach that uses the

internal states of the LLM, which may not be available through API calls, and requires labelled data for supervised training. Another recent approach is self-evaluation (Kadavath et al., 2022), where an LLM is prompted to evaluate its previous prediction, e.g., to predict the probability that its generated response/answer is true.

### 2.2 Sequence Level Uncertainty Estimation

Token probabilities have been used as an indication of model certainty. For example, OpenAI's GPT-3 web interface allows users to display token probabilities (as shown in Figure 2), and further uncertainty estimation approaches based on aleatoric and epistemic uncertainty have been studied for autoregressive generation (Xiao and Wang, 2021; Malinin and Gales, 2021). Additionally, conditional language model scores have been used to evaluate properties of texts (Yuan et al., 2021; Fu et al., 2023). Recently, semantic uncertainty has been proposed to address uncertainty in free-form generation tasks where probabilities are attached to concepts instead of tokens (Kuhn et al., 2023).

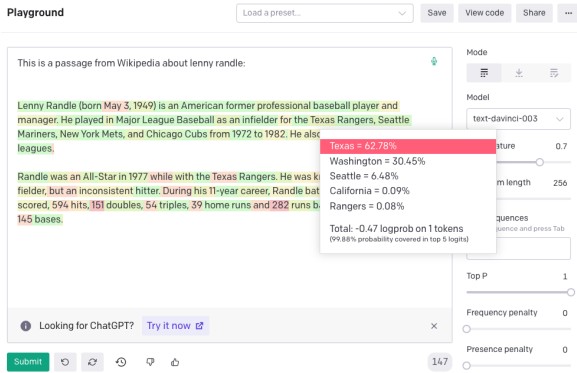

**Figure 2:** Example of OpenAI's GPT-3 web interface with output token-level probabilities displayed.

### 2.3 Fact Verification

Existing fact-verification approaches follow a multi-stage pipeline of claim detection, evidence retrieval and verdict prediction (Guo et al., 2022; Zhong et al., 2020). Such methods, however, require access to external databases and can have considerable inference costs.

## 3 Grey-Box Factuality Assessment

This section will introduce methods that can be used to determine the factuality of LLM responses in a zero-resource setting when one has full access

to output distributions.[2] We will use 'factual' to define when statements are grounded in valid information, i.e. when hallucinations are avoided, and 'zero-resource' when no external database is used.

## 3.1 Uncertainty-based Assessment

To understand how the factuality of a generated response can be determined in a zero-resource setting, we consider LLM pre-training. During pre-training, the model is trained with next-word prediction over massive corpora of textual data. This gives the model a strong understanding of language (Jawahar et al., 2019; Raffel et al., 2020), powerful contextual reasoning (Zhang et al., 2020), as well as world knowledge (Liusie et al., 2023). Consider the input "Lionel Messi is a _". Since Messi is a world-famous athlete who may have appeared multiple times in pre-training, the LLM is likely to know who Messi is. Therefore given the context, the token "footballer" may be assigned a high probability while other professions such as "carpenter" may be considered improbable. However, for a different input such as "John Smith is a _", the system will be unsure of the continuation which may result in a flat probability distribution. During inference, this is likely to lead to a non-factual word being generated.

This insight allows us to understand the connection between uncertainty metrics and factuality. Factual sentences are likely to contain tokens with higher likelihood and lower entropy, while hallucinations are likely to come from positions with flat probability distributions with high uncertainty.

## Token-level Probability

Given the LLM's response $R$, let $i$ denote the $i$-th sentence in $R$, $j$ denote the $j$-th token in the $i$-th sentence, $J$ is the number of tokens in the sentence, and $p_{ij}$ be the probability of the word generated by the LLM at the $j$-th token of the $i$-th sentence. Two probability metrics are used:

$$\text{Avg}(-\log p) = -\frac{1}{J} \sum_j \log p_{ij}$$
$$\text{Max}(-\log p) = \max_j \left(-\log p_{ij}\right)$$

$\text{Max}(-\log p)$ measures the sentence's likelihood by assessing the *least* likely token in the sentence.

---

[2]Alternate white-box approaches such as that of Azaria and Mitchell (2023) require access to full internal states, and is less practical and so not considered in this work.

## Entropy

The entropy of the output distribution is:

$$\mathcal{H}_{ij} = -\sum_{\tilde{w} \in \mathcal{W}} p_{ij}(\tilde{w}) \log p_{ij}(\tilde{w})$$

where $p_{ij}(\tilde{w})$ is the probability of the word $\tilde{w}$ being generated at the $j$-th token of the $i$-th sentence, and $\mathcal{W}$ is the set of all possible words in the vocabulary. Similar to the probability-based metrics, two entropy-based metrics are used:

$$\text{Avg}(\mathcal{H}) = \frac{1}{J} \sum_j \mathcal{H}_{ij}; \quad \text{Max}(\mathcal{H}) = \max_j \left(\mathcal{H}_{ij}\right)$$

## 4 Black-Box Factuality Assessment

A drawback of grey-box methods is that they require output token-level probabilities. Though this may seem a reasonable requirement, for massive LLMs only available through limited API calls, such token-level information may not be available (such as with ChatGPT). Therefore, we consider black-box approaches which remain applicable even when only text-based responses are available.

## Proxy LLMs

A simple approach to approximate the grey-box approaches is by using a proxy LLM, i.e. another LLM that we have full access to, such as LLaMA (Touvron et al., 2023). A proxy LLM can be used to approximate the output token-level probabilities of the black-box LLM generating the text. In the next section, we propose SelfCheckGPT, which is also a black-box approach.

## 5 SelfCheckGPT

SelfCheckGPT is our proposed black-box zero-resource hallucination detection scheme, which operates by comparing multiple sampled responses and measuring consistency.

**Notation**: Let $R$ refer to an LLM response drawn from a given user query. SelfCheckGPT draws a further $N$ stochastic LLM response samples $\{S^1, S^2, ..., S^n, ..., S^N\}$ using the same query, and then measures the consistency between the response and the stochastic samples. We design SelfCheckGPT to predict the hallucination score of the $i$-th sentence, $\mathcal{S}(i)$, such that $\mathcal{S}(i) \in [0.0, 1.0]$, where $\mathcal{S}(i) \to 0.0$ if the $i$-th sentence is grounded in valid information and $\mathcal{S}(i) \to 1.0$ if the $i$-th sen-

tence is hallucinated.[3] The following subsections will describe each of the SelfCheckGPT variants.

## 5.1 SelfCheckGPT with BERTScore

Let $\mathcal{B}(.,.)$ denote the BERTScore between two sentences. SelfCheckGPT with BERTScore finds the average BERTScore of the $i$-th sentence with the most similar sentence from each drawn sample:

$$\mathcal{S}_{\text{BERT}}(i) = 1 - \frac{1}{N} \sum_{n=1}^{N} \max_{k} \left( \mathcal{B}(r_i, s_k^n) \right) \quad (1)$$

where $r_i$ represents the $i$-th sentence in $R$ and $s_k^n$ represents the $k$-th sentence in the $n$-th sample $S^n$. This way if the information in a sentence appears in many drawn samples, one may assume that the information is factual, whereas if the statement appears in no other sample, it is likely a hallucination. In this work, RoBERTa-Large (Liu et al., 2019) is used as the backbone of BERTScore.

## 5.2 SelfCheckGPT with Question Answering

We also consider using the automatic multiple-choice question answering generation (MQAG) framework (Manakul et al., 2023) to measure consistency for SelfCheckGPT. MQAG assesses consistency by generating multiple-choice questions over the main generated response, which an independent answering system can attempt to answer while conditioned on the other sampled responses. If questions on consistent information are queried, the answering system is expected to predict similar answers. MQAG consists of two stages: question generation G and question answering A. For the sentence $r_i$ in the response $R$, we draw questions $q$ and options $\mathbf{o}$:

$$q, \mathbf{o} \sim P_{\mathsf{G}}(q, \mathbf{o} | r_i, R) \quad (2)$$

The answering stage A selects the answers:

$$a_R = \underset{k}{\arg\max} \left[ P_{\mathsf{A}}(o_k | q, R, \mathbf{o}) \right] \quad (3)$$

$$a_{S^n} = \underset{k}{\arg\max} \left[ P_{\mathsf{A}}(o_k | q, S^n, \mathbf{o}) \right] \quad (4)$$

We compare whether $a_R$ is equal to $a_{S^n}$ for each sample in $\{S^1, ..., S^N\}$, yielding #matches $N_{\mathtt{m}}$ and #not-matches $N_{\mathtt{n}}$. A simple inconsistency score for the $i$-th sentence and question $q$ based on the match/not-match counts is defined: $\mathcal{S}_{\text{QA}}(i, q) =$

$\frac{N_{\mathtt{n}}}{N_{\mathtt{m}} + N_{\mathtt{n}}}$. To take into account the answerability of generated questions, we show in Appendix B that we can modify the inconsistency score by applying soft-counting, resulting in:

$$\mathcal{S}_{\text{QA}}(i, q) = \frac{\gamma_2^{N'_{\mathtt{n}}}}{\gamma_1^{N'_{\mathtt{m}}} + \gamma_2^{N'_{\mathtt{n}}}} \quad (5)$$

where $N'_{\mathtt{m}}$ = the effective match count, $N'_{\mathtt{n}}$ = the effective mismatch count, with $\gamma_1$ and $\gamma_2$ defined in Appendix B.1. Ultimately, SelfCheckGPT with QA is the average of inconsistency scores across $q$,

$$\mathcal{S}_{\text{QA}}(i) = \mathbb{E}_q \left[ \mathcal{S}_{\text{QA}}(i, q) \right] \quad (6)$$

## 5.3 SelfCheckGPT with n-gram

Given samples $\{S^1, ..., S^N\}$ generated by an LLM, one can use the samples to create a new language model that approximates the LLM. In the limit as $N$ gets sufficiently large, the new language model will converge to the LLM that generated the responses. We can therefore approximate the LLM's token probabilities using the new language model.

In practice, due to time and/or cost constraints, there can only be a limited number of samples $N$. Consequently, we train a simple $n$-gram model using the samples $\{S^1, ..., S^N\}$ as well as the main response $R$ (which is assessed), where we note that including $R$ can be considered as a smoothing method where the count of each token in $R$ is increased by 1. We then compute the average of the log-probabilities of the sentence in response $R$,

$$\mathcal{S}_{n\text{-gram}}^{\text{Avg}}(i) = -\frac{1}{J} \sum_j \log \tilde{p}_{ij} \quad (7)$$

where $\tilde{p}_{ij}$ is the probability (of the $j$-th token of the $i$-th sentence) computed using the $n$-gram model. Similar to the grey-box approach, we can also use the maximum of the negative log probabilities,

$$\mathcal{S}_{n\text{-gram}}^{\text{Max}}(i) = \max_j \left( -\log \tilde{p}_{ij} \right) \quad (8)$$

## 5.4 SelfCheckGPT with NLI

Natural Language Inference (NLI) determines whether a hypothesis follows a premise, classified into either entailment/neutral/contradiction. NLI measures have been used to measure faithfulness in summarization, where Maynez et al. (2020) use a textual entailment classifier trained on MNLI (Williams et al., 2018) to determine if a summary contradicts a context or not. Inspired by NLI-based

---

[3]With the exception of SelfCheckGPT with $n$-gram as the score of the $n$-gram language model is not bounded.

summary assessment, we consider using the NLI contradiction score as a SelfCheckGPT score.

For SelfCheck-NLI, we use DeBERTa-v3-large (He et al., 2023) fine-tuned to MNLI as the NLI model. The input for NLI classifiers is typically the `premise` concatenated to the `hypothesis`, which for our methodology is the sampled passage $S^n$ concatenated to the sentence to be assessed $r_i$. Only the logits associated with the 'entailment' and 'contradiction' classes are considered,

$$P(\text{contradict}|r_i, S^n) = \frac{\exp(z_c)}{\exp(z_e) + \exp(z_c)} \quad (9)$$

where $z_e$ and $z_c$ are the logits of the 'entailment' and 'contradiction' classes, respectively. This normalization ignores the neutral class and ensures that the probability is bounded between 0.0 and 1.0. The SelfCheckGPT with NLI score for each sample $S^n$ is then defined as,

$$\mathcal{S}_{\text{NLI}}(i) = \frac{1}{N} \sum_{n=1}^{N} P(\text{contradict}|r_i, S^n) \quad (10)$$

### 5.5 SelfCheckGPT with Prompt

LLMs have recently been shown to be effective in assessing information consistency between a document and its summary in zero-shot settings (Luo et al., 2023). Thus, we query an LLM to assess whether the $i$-th sentence is supported by sample $S^n$ (as the context) using the following prompt.

```
------------------------------------------------
Context: {}
Sentence: {}
Is the sentence supported by the context above?
Answer Yes or No:
------------------------------------------------
```

Initial investigation showed that GPT-3 (text-davinci-003) will output either `Yes` or `No` 98% of the time, while any remaining outputs can be set to `N/A`. The output from prompting when comparing the $i$-th sentence against sample $S^n$ is converted to score $x_i^n$ through the mapping {`Yes`: 0.0, `No`: 1.0, `N/A`: 0.5}. The final inconsistency score is then calculated as:

$$\mathcal{S}_{\text{Prompt}}(i) = \frac{1}{N} \sum_{n=1}^{N} x_i^n \quad (11)$$

SelfCheckGPT-Prompt is illustrated in Figure 1. Note that our initial investigations found that less capable models such as GPT-3 (text-curie-001) or LLaMA failed to effectively perform consistency assessment via such prompting.

## 6 Data and Annotation

As, currently, there are no standard hallucination detection datasets available, we evaluate our hallucination detection approaches by 1) generating synthetic Wikipedia articles using GPT-3 on the individuals/concepts from the WikiBio dataset (Lebret et al., 2016); 2) manually annotating the factuality of the passage at a sentence level; 3) evaluating the system's ability to detect hallucinations.

WikiBio is a dataset where each input contains the first paragraph (along with tabular information) of Wikipedia articles of a specific concept. We rank the WikiBio test set in terms of paragraph length and randomly sample 238 articles from the top 20% of longest articles (to ensure no very obscure concept is selected). GPT-3 (text-davinci-003) is then used to generate Wikipedia articles on a concept, using the prompt `"This is a Wikipedia passage about {concept}:"`. Table 1 provides the statistics of GPT-3 generated passages.

| #Passages | #Sentences | #Tokens/passage |
|-----------|-----------|-----------------|
| 238 | 1908 | 184.7±36.9 |

**Table 1:** The statistics of **WikiBio GPT-3 dataset** where the number of tokens is based on the OpenAI GPT-2 tokenizer.

We then annotate the sentences of the generated passages using the guidelines shown in Figure 3 such that each sentence is classified as either:

- **Major Inaccurate** (Non-Factual, **1**): The sentence is entirely hallucinated, i.e. the sentence is unrelated to the topic.

- **Minor Inaccurate** (Non-Factual, **0.5**): The sentence consists of some non-factual information, but the sentence is related to the topic.

- **Accurate** (Factual, **0**): The information presented in the sentence is accurate.

Of the 1908 annotated sentences, 761 (39.9%) of the sentences were labelled major-inaccurate, 631 (33.1%) minor-inaccurate, and 516 (27.0%) accurate. 201 sentences in the dataset had annotations from two different annotators. To obtain a single label for this subset, if both annotators agree, then the agreed label is used. However, if there is disagreement, then the worse-case label is selected (e.g., {minor inaccurate, major inaccurate} is mapped to major inaccurate). The inter-annotator agreement, as measured by Cohen's $\kappa$ (Cohen, 1960), has $\kappa$

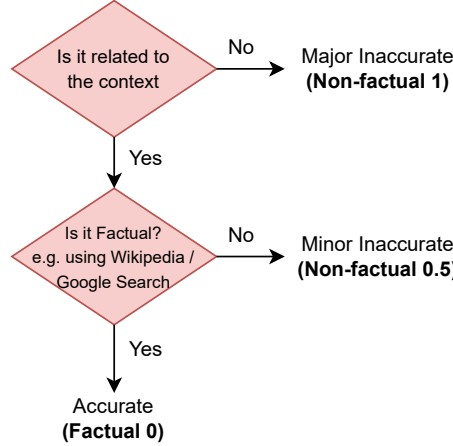

**Figure 3:** Flowchart of our annotation process

values of 0.595 and 0.748, indicating *moderate* and *substantial* agreement (Viera et al., 2005) for the 3-class and 2-class scenarios, respectively.[4]

Furthermore, passage-level scores are obtained by averaging the sentence-level labels in each passage. The distribution of passage-level scores is shown in Figure 4, where we observe a large peak at +1.0. We refer to the points at this peak as *total hallucination*, which occurs when the information of the response is unrelated to the real concept and is entirely fabricated by the LLM.

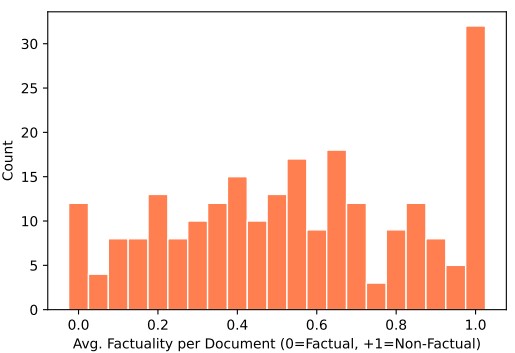

**Figure 4:** Document factuality scores histogram plot

## 7 Experiments

The generative LLM used to generate passages for our dataset is **GPT-3** (text-davinci-003), the state-of-the-art system at the time of creating and annotating the dataset. To obtain the main response, we set the temperature to 0.0 and use standard beam search decoding. For the stochastically generated samples, we set the temperature to 1.0 and generate

[4]3-class refers to when selecting between accurate, minor inaccurate, major inaccurate. 2-class refers to when minor/major inaccuracies are combined into one label.

$N$=20 samples. For the proxy LLM approach, we use LLaMA (Touvron et al., 2023), one of the best-performing open-source LLMs currently available. For SelfCheckGPT-Prompt, we consider both GPT-3 (which is the same LLM that is used to generate passages) as well as the newly released ChatGPT (gpt-3.5-turbo). More details about the systems in SelfCheckGPT and results using other proxy LLMs can be found in the appendix.

### 7.1 Sentence-level Hallucination Detection

First, we investigate whether our hallucination detection methods can identify the factuality of sentences. In detecting non-factual sentences, both major-inaccurate labels and minor-inaccurate labels are grouped together into the *non-factual* class, while the *factual* class refers to accurate sentences. In addition, we consider a more challenging task of detecting major-inaccurate sentences in passages that are *not* total hallucination passages, which we refer to as *non-factual*[*].[5] Figure 5 and Table 2 show the performance of our approaches, where the following observations can be made:

**1) LLM's probabilities $p$ correlate well with factuality**. Our results show that probability measures (from the LLM generating the texts) are strong baselines for assessing factuality. Factual sentences can be identified with an AUC-PR of 53.97, significantly better than the random baseline of 27.04, with the AUC-PR for hallucination detection also increasing from 72.96 to 83.21. This supports the hypothesis that when the LLMs are uncertain about generated information, generated tokens often have higher uncertainty, paving a promising direction for hallucination detection approaches. Also, the probability $p$ measure performs better than the entropy $\mathcal{H}$ measure of top-5 tokens.

**2) Proxy LLM perform noticeably worse than LLM (GPT-3)**. The results of proxy LLM (based on LLaMA) show that the entropy $\mathcal{H}$ measures outperform the probability measures. This suggests that using richer uncertainty information can improve factuality/hallucination detection performance, and that previously the entropy of top-5 tokens is likely to be insufficient. In addition, when using other proxy LLMs such as GPT-NeoX or OPT-30B, the performance is near that of the random baseline. We believe this poor performance occurs as different LLMs have different generating patterns, and so even common tokens may have a

[5]There are 206 non-factual[*] passages (1632 sentences).

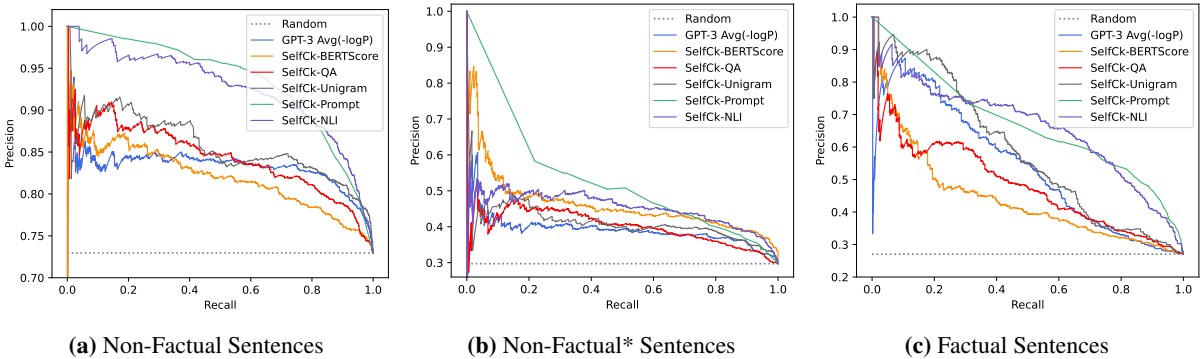

| (a) Non-Factual Sentences | (b) Non-Factual* Sentences | (c) Factual Sentences |

**Figure 5:** PR-Curve of detecting non-factual and factual *sentences* in the GPT-3 generated WikiBio passages.

| Method | Sentence-level (AUC-PR) | | | Passage-level (Corr.) | |
|---|---|---|---|---|---|
| | NonFact | NonFact* | Factual | Pearson | Spearman |
| Random | 72.96 | 29.72 | 27.04 | - | - |
| GPT-3 (`text-davinci-003`)'s probabilities (*LLM, grey-box*) | | | | | |
| Avg($-\log p$) | 83.21 | 38.89 | 53.97 | 57.04 | 53.93 |
| Avg($\mathcal{H}$)[†] | 80.73 | 37.09 | 52.07 | 55.52 | 50.87 |
| Max($-\log p$) | 87.51 | 35.88 | 50.46 | 57.83 | 55.69 |
| Max($\mathcal{H}$)[†] | 85.75 | 32.43 | 50.27 | 52.48 | 49.55 |
| LLaMA-30B's probabilities (*Proxy LLM, black-box*) | | | | | |
| Avg($-\log p$) | 75.43 | 30.32 | 41.29 | 21.72 | 20.20 |
| Avg($\mathcal{H}$) | 80.80 | 39.01 | 42.97 | 33.80 | 39.49 |
| Max($-\log p$) | 74.01 | 27.14 | 31.08 | -22.83 | -22.71 |
| Max($\mathcal{H}$) | 80.92 | 37.32 | 37.90 | 35.57 | 38.94 |
| **SelfCheckGPT** (*black-box*) | | | | | |
| w/ BERTScore | 81.96 | 45.96 | 44.23 | 58.18 | 55.90 |
| w/ QA | 84.26 | 40.06 | 48.14 | 61.07 | 59.29 |
| w/ Unigram (max) | 85.63 | 41.04 | 58.47 | 64.71 | 64.91 |
| w/ NLI | 92.50 | 45.17 | 66.08 | 74.14 | 73.78 |
| w/ Prompt | **93.42** | **53.19** | **67.09** | **78.32** | **78.30** |

**Table 2:** AUC-PR for sentence-level detection tasks. Passage-level ranking performances are measured by Pearson correlation coefficient and Spearman's rank correlation coefficient w.r.t. human judgements. The results of other proxy LLMs, in addition to LLaMA, can be found in the appendix. [†]GPT-3 API returns the top-5 tokens' probabilities, which are used to compute entropy.

low probability in situations where the response is dissimilar to the generation style of the proxy LLM. We note that a weighted conditional LM score such as BARTScore (Yuan et al., 2021) could be incorporated in future investigations.

**3) SelfCheckGPT outperforms grey-box approaches**. It can be seen that SelfCheckGPT-Prompt *considerably* outperforms the grey-box approaches (including GPT-3's output probabilities) as well as other black-box approaches. Even other variants of SelfCheckGPT, including BERTScore, QA, and $n$-gram, outperform the grey-box approaches in most setups. Interestingly, despite being the least computationally expensive method, SelfCheckGPT with unigram (max) works well

across different setups. Essentially, when assessing a sentence, this method picks up the token with the *lowest* occurrence given all the samples. This suggests that if a token only appears a few times (or once) within the generated samples ($N$=20), it is likely non-factual.

**4) SelfCheckGPT with $n$-gram**. When investigating the $n$-gram performance from 1-gram to 5-gram, the results show that simply finding the least likely token/$n$-gram is more effective than computing the average $n$-gram score of the sentence, details in appendix Table 7. Additionally, as $n$ increases, the performance of SelfCheckGPT with $n$-gram (max) drops.

**5) SelfCheckGPT with NLI**. The NLI-based

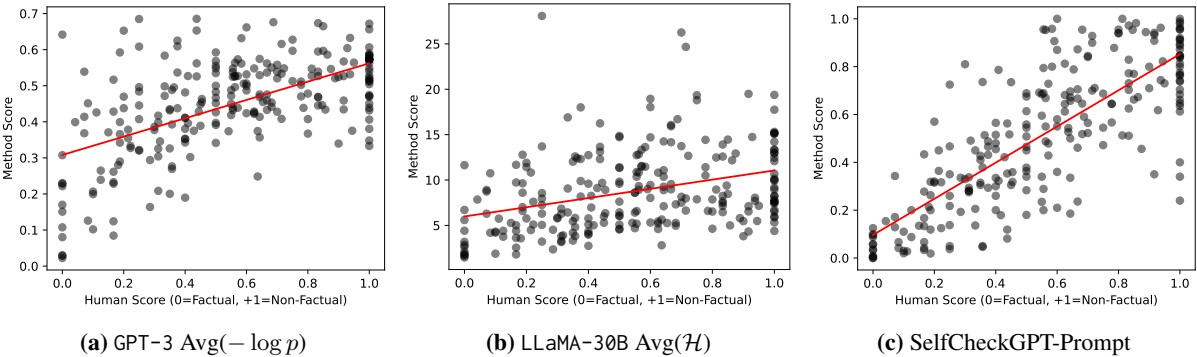

**(a)** GPT-3 Avg$(-\log p)$      **(b)** LLaMA-30B Avg$(\mathcal{H})$      **(c)** SelfCheckGPT-Prompt

**Figure 6:** Scatter plot of passage-level scores where Y-axis = Method scores, X-axis = Human scores. Correlations are reported in Table 2. The scatter plots of other SelfCheckGPT variants are provided in Figure 10 in the appendix.

method outperforms all black-box and grey-box baselines, and its performance is close to the performance of the Prompt method. As SelfCheckGPT with Prompt can be computationally heavy, SelfCheckGPT with NLI could be the most practical method as it provides a good trade-off between performance and computation.

## 7.2 Passage-level Factuality Ranking

Previous results demonstrate that SelfCheckGPT is an effective approach for predicting sentence-level factuality. An additional consideration is whether SelfCheckGPT can also be used to determine the overall factuality of passages. Passage-level factuality scores are calculated by averaging the sentence-level scores over all sentences.

$$\mathcal{S}_{\text{passage}} = \frac{1}{|R|} \sum_i \mathcal{S}(i) \qquad (12)$$

where $\mathcal{S}(i)$ is the sentence-level score, and $|R|$ is the number of sentences in the passage. Since human judgement is somewhat subjective, averaging the sentence-level labels would lead to ground truths with less noise. Note that for Avg$(-\log p)$ and Avg$(\mathcal{H})$, we compute the average over all tokens in a passage. Whereas for Max$(-\log p)$ and Max$(\mathcal{H})$, we first take the maximum operation over tokens at the sentence level, and we then average over all sentences following Equation 12.

Our results in Table 2 and Figure 6 show that all SelfCheckGPT methods correlate far better with human judgements than the other baselines, including the grey-box probability and entropy methods. SelfCheckGPT-Prompt is the best-performing method, achieving the highest Pearson correlation of 78.32. Unsurprisingly, the proxy LLM approach again achieves considerably lower correlations.

## 7.3 Ablation Studies

### External Knowledge (instead of SelfCheck)

If external knowledge is available, one can measure the informational consistency between the LLM response and the information source. In this experiment, we use the first paragraph of each concept that is available in WikiBio.[6]

| Method | Sent-lvl AUC-PR | | | Passage-lvl | |
|---|---|---|---|---|---|
| | NoFac | NoFac* | Fact | Pear. | Spear. |
| SelfCk-BERT | 81.96 | 45.96 | 44.23 | 58.18 | 55.90 |
| WikiBio+BERT | 81.32 | 40.62 | 49.15 | 58.71 | 55.80 |
| SelfCk-QA | 84.26 | 40.06 | 48.14 | 61.07 | 59.29 |
| WikiBio+QA | 84.18 | 45.40 | 52.03 | 57.26 | 53.62 |
| SelfCk-1gm | 85.63 | 41.04 | 58.47 | 64.71 | 64.91 |
| WikiBio+1gm | 80.43 | 31.47 | 40.53 | 28.67 | 26.70 |
| SelfCk-NLI | 92.50 | 45.17 | 66.08 | 74.14 | 73.78 |
| WikiBio+NLI | 91.18 | 48.14 | 71.61 | 78.84 | 80.00 |
| SelfCk-Prompt | 93.42 | 53.19 | 67.09 | 78.32 | 78.30 |
| WikiBio+Prompt | 93.59 | 65.26 | 73.11 | 85.90 | 86.11 |

**Table 3:** The performance when using SelfCheckGPT samples versus external stored knowledge.

Our findings in Table 3 show the following. First, SelfCheckGPT with BERTScore/QA, using self-samples, can yield comparable or even better performance than when using the reference passage. Second, SelfCheckGPT with $n$-gram shows a large performance drop when using the WikiBio passages instead of self-samples. This failure is attributed to the fact that the WikiBio reference text alone is not sufficient to train an $n$-gram model. Third, in contrast, SelfCheckGPT with NLI/Prompt can benefit considerably when access to retrieved information is available. Nevertheless, in practice,

---

[6]This method is no longer zero-resource as it requires retrieving relevant knowledge from external data.

it is infeasible to have an external database for every possible use case of LLM generation.

**The Impact of the Number of Samples**

Although sample-based methods are expected to perform better when more samples are drawn, this has higher computational costs. Thus, we investigate performance as the number of samples is varied. Our results in Figure 7 show that the performance of SelfCheckGPT increases smoothly as more samples are used, with diminishing gains as more samples are generated. SelfCheckGPT with $n$-gram requires the highest number of samples before its performance reaches a plateau.

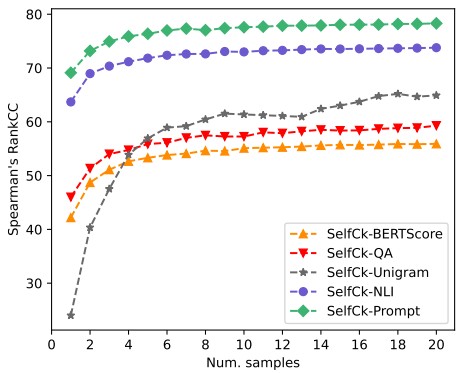

**Figure 7:** The performance of SelfCheckGPT methods on ranking passages (Spearman's) versus the number of samples.

**The Choice of LLM for SelfCheckGPT-Prompt**

We investigate whether the LLM generating the text can self-check its own text. We conduct this ablation using a reduced set of the samples ($N$=4).

| Text-Gen | SelfCk-Prompt | $N$ | Pear. | Spear. |
|---|---|---|---|---|
| GPT-3 | ChatGPT | 20 | 78.32 | 78.30 |
| GPT-3 | ChatGPT | 4 | 76.47 | 76.41 |
| GPT-3 | GPT-3 | 4 | 73.11 | 74.69 |
| [†]SelfCheck w/ unigram (max) | | 20 | 64.71 | 64.91 |
| [†]SelfCheck w/ NLI | | 20 | 74.14 | 73.78 |

**Table 4:** Comparison of GPT-3 (text-davinci-003) and ChatGPT (gpt-3.5.turbo) as the prompt-based text evaluator in SelfCheckGPT-Prompt. [†]Taken from Table 2 for comparison.

The results in Table 4 show that GPT-3 can self-check its own text, and is better than the unigram method even when using only 4 samples. However, ChatGPT shows a slight improvement over GPT-3 in evaluating whether the sentence is supported by the context. More details are in Appendix C.

# 8 Conclusions

This paper is the first work to consider the task of hallucination detection for general large language model responses. We propose SelfCheckGPT, a zero-resource approach that is applicable to any black-box LLM without the need for external resources, and demonstrate the efficacy of our method. SelfCheckGPT outperforms a range of considered grey-box and black-box baseline detection methods at both the sentence and passage levels, and we further release an annotated dataset for GPT-3 hallucination detection with sentence-level factuality labels.

## Limitations

In this study, the 238 GPT-3 generated texts were predominantly passages about individuals in the WikiBio dataset. To further investigate the nature of LLM's hallucination, this study could be extended to a wider range of concepts, e.g., to also consider generated texts about locations and objects. Further, this work considers factuality at the sentence level, but we note that a single sentence may consist of both factual and non-factual information. For example, the following work by Min et al. (2023) considers a fine-grained factuality evaluation by decomposing sentences into atomic facts. Finally, SelfCheckGPT with Prompt, which was convincingly the best selfcheck method, is quite computationally heavy. This might lead to impractical computational costs, which could be addressed in future work to be made more efficient.

## Ethics Statement

As this work addresses the issue of LLM's hallucination, we note that if hallucinated contents are not detected, they could lead to misinformation.

## Acknowledgments

This work is supported by Cambridge University Press & Assessment (CUP&A), a department of The Chancellor, Masters, and Scholars of the University of Cambridge, and the Cambridge Commonwealth, European & International Trust. We would like to thank the anonymous reviewers for their helpful comments.

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

## A Models and Implementation

### A.1 Entropy

The entropy of the output distribution is implemented as follows,

$$\mathcal{H}_{ij} = 2^{-\sum_{\tilde{w} \in \mathcal{W}} p_{ij}(\tilde{w}) \log_2 p_{ij}(\tilde{w})} \qquad (13)$$

where $\mathcal{W}$ is the set of all possible words in the vocabulary.

### A.2 Proxy LLMs

The proxy LLMs considered are LLaMA-{7B, 13B, 30B} (Touvron et al., 2023), OPT-{125m, 1.3B, 13B, 30B} (Zhang et al., 2022), GPT-J-6B (Wang and Komatsuzaki, 2021) and GPT-NeoX-20B (Black et al., 2022).

### A.3 SelfCheckGPT's Systems

**Question Answering**: The generation systems G1 and G2 are T5-Large fine-tuned to SQuAD (Rajpurkar et al., 2016) and RACE (Lai et al., 2017), respectively. The answering system A is Longformer (Beltagy et al., 2020) fine-tuned to the RACE dataset. The answerability system U is also Longformer, but fine-tuned to SQuAD2.0.

**LLM for Prompting**: We consider two LLMs, GPT-3 (text-davinci-003) and ChatGPT (gpt-3.5-turbo) We note that during the data creation and annotation, GPT-3 (text-davinci-003) was the state-of-the-art LLM available; hence, GPT-3 was used as the main LLM generating WikiBio passages.

## B SelfCheckGPT with QA

Previous work showed that implementing question generation (in Equation 2) with two generators (G1 generates the question and associated answer, and G2 generates distractors) yields higher-quality distractors (Manakul et al., 2023). Thus, a two-stage generation is adopted in this work as follows:

$$q, a \sim P_{\text{G1}}(q, a|r_i); \quad \mathbf{o}_{\backslash a} \sim P_{\text{G2}}(\mathbf{o}_{\backslash a}|q, a, R) \qquad (14)$$

where $\mathbf{o} = \{a, \mathbf{o}_{\backslash a}\} = \{o_1, ..., o_4\}$. In addition, to filter out bad (unanswerable) questions, we define an answerability score (Raina and Gales, 2022):

$$\alpha = P_{\text{U}}(\text{answerable}|q, \text{context}) \qquad (15)$$

where the context is either the response $R$ or sampled passages $S^n$, and $\alpha \to 0.0$ for unanswerable and $\alpha \to 1.0$ for answerable. We use $\alpha$ to filter out

unanswerable questions which have $\alpha$ lower than a threshold. Next, we derive how Bayes' theorem can be applied to take into account the number of answerable/unanswerable questions.

### B.1 SelfCheckGPT-QA with Bayes

Let $P(\text{F})$ denote the probability of the $i$-th sentence being non-factual, and $P(\text{T})$ denote the probability of the $i$-th sentence being factual. For a question $q$, the probability of $i$-th sentence being non-factual given a set of matched answers $L_{\text{m}}$ and a set of not-matched answers $L_{\text{n}}$ is:

$$\begin{aligned} & P(\text{F}|L_{\text{m}}, L_{\text{n}}) \\ &= \frac{P(L_{\text{m}}, L_{\text{n}}|\text{F})P(\text{F})}{P(L_{\text{m}}, L_{\text{n}}|\text{F})P(\text{F}) + P(L_{\text{m}}, L_{\text{n}}|\text{T})P(\text{T})} \\ &= \frac{P(L_{\text{m}}, L_{\text{n}}|\text{F})}{P(L_{\text{m}}, L_{\text{n}}|\text{F}) + P(L_{\text{m}}, L_{\text{n}}|\text{T})} \end{aligned} \qquad (16)$$

where we assume the sentence is equally likely to be False or True, i.e. $P(\text{F}) = P(\text{T})$. The probability of observing $L_{\text{m}}, L_{\text{n}}$ when the sentence is False (non-factual):

$$\begin{aligned} & P(L_{\text{m}}, L_{\text{n}}|\text{F}) \\ &= \prod_{a \in L_{\text{m}}} P(a = a_R|F) \prod_{a' \in L_{\text{n}}} P(a' \neq a_R|F) \\ &= (1 - \beta_1)^{N_{\text{m}}}(\beta_1)^{N_{\text{n}}} \end{aligned} \qquad (17)$$

and probability of observing $L_{\text{m}}, L_{\text{n}}$ when the sentence is True (factual):

$$\begin{aligned} & P(L_{\text{m}}, L_{\text{n}}|\text{T}) \\ &= \prod_{a \in L_{\text{m}}} P(a = a_r|T) \prod_{a' \in L_{\text{n}}} P(a' \neq a_r|T) \\ &= (\beta_2)^{N_{\text{m}}}(1 - \beta_2)^{N_{\text{n}}} \end{aligned} \qquad (18)$$

where $N_{\text{m}}$ and $N_{\text{n}}$ are the number of matched answers and the number of not-matched answers, respectively. Hence, we can simplify Equation 16:

$$P(\text{F}|L_{\text{m}}, L_{\text{n}}) = \frac{\gamma_2^{N_{\text{n}}}}{\gamma_1^{N_{\text{m}}} + \gamma_2^{N_{\text{n}}}} \qquad (19)$$

where $\gamma_1 = \frac{\beta_2}{1 - \beta_1}$ and $\gamma_2 = \frac{\beta_1}{1 - \beta_2}$. Lastly, instead of rejecting samples having an answerability score below a threshold,[7] we find empirically that soft-counting (defined below) improves the detection performance. We set both $\beta_1$ and $\beta_2$ to 0.8.

---

[7] $\alpha$ is between 0.0 (unanswerable) and 1.0 (answerable). Standard-counting $N_{\text{m}}$ and $N_{\text{n}}$ can be considered as a special case of soft-counting where $\alpha$ is set to 1.0 if $\alpha$ is greater than the answerability threshold and otherwise $\alpha$ is 0.0.

$$N'_{\mathtt{m}} = \sum_{n \text{ s.t. } a_n \in L_{\mathtt{m}}} \alpha_n; \quad N'_{\mathtt{n}} = \sum_{n \text{ s.t. } a_n \in L_{\mathtt{n}}} \alpha_n \quad (20)$$

where $\alpha_n = P_{\mathtt{U}}(\text{answerable}|q, S^n)$. Therefore, the SelfCheckGPT with QA score, $\mathcal{S}_{\mathrm{QA}}$, is:

$$\mathcal{S}_{\mathrm{QA}} = P(\mathrm{F}|L_{\mathtt{m}}, L_{\mathtt{n}}) = \frac{\gamma_2^{N'_{\mathtt{n}}}}{\gamma_1^{N'_{\mathtt{m}}} + \gamma_2^{N'_{\mathtt{n}}}} \quad (21)$$

In Table 5, we show empically that applying Bayes' theorem and soft counting $\alpha$ (in Equation 20) improves the performance of the SelfCheckGPT with QA method.

| Varaint | Sentence-lvl | | | Passage-lvl | |
|---|---|---|---|---|---|
| | NoF | NoF* | Fact | PCC | SCC |
| SimpleCount | 83.97 | 40.07 | 47.78 | 57.39 | 55.15 |
| + Bayes | 83.04 | 38.58 | 47.41 | 56.43 | 55.03 |
| + Bayes + $\alpha$ | 84.26 | 40.06 | 48.14 | 61.07 | 59.29 |

**Table 5:** Performance of SelfCheckGPT-QA's variants.

## C  SelfCheckGPT with Prompt

We use the prompt template provided in the main text (in Section 5.5) for both GPT-3 (text-davinci-003) and ChatGPT (gpt-3.5-turbo). For ChatGPT, a standard system message `"You are a helpful assistant."` is used in setting up the system.

At the time of conducting experiments, the API costs per 1,000 tokens are \$0.020 for GPT-3 and \$0.002 for ChatGPT. The estimated costs for running the models to answer `Yes`/`No` on all 1908 sentences and 20 samples are around \$200 for GPT-3 and \$20 for ChatGPT. Given the cost, we conduct the experiments on 4 samples when performing the ablation about LLM choice for SelfCheckGPT-Prompt (Section 7.3). Table 6 shows the breakdown of predictions made by GPT-3 and ChatGPT.

| GPT-3 ╲ ChatGPT | Yes | No |
|---|---|---|
| Yes | 3179 | 1038 |
| No | 367 | 3048 |

**Table 6:** Breakdown of predictions made by GPT-3/ChatGPT when prompted to answer `Yes`(supported)/`No`(not-supported).

## D  Additional Experimental Results

Here, we provide experimental results that are complementary to those presented in the main paper.

| $n$-gram | Sent-lvl AUC-PR | | | Passage-lvl | |
|---|---|---|---|---|---|
| | NoFac | NoFac* | Fact | Pear. | Spear. |
| Avg($-\log p$) | | | | | |
| 1-gram | 81.52 | 40.33 | 41.76 | 40.68 | 39.22 |
| 2-gram | 82.94 | 44.38 | 52.81 | 58.84 | 58.11 |
| 3-gram | 83.56 | 44.64 | 53.99 | 62.21 | 63.00 |
| 4-gram | 83.80 | 43.55 | 54.25 | 61.98 | 63.64 |
| 5-gram | 83.45 | 42.31 | 53.98 | 60.68 | 62.96 |
| Max($-\log p$) | | | | | |
| 1-gram | 85.63 | 41.04 | 58.47 | 64.71 | 64.91 |
| 2-gram | 85.26 | 39.29 | 58.29 | 62.48 | 66.04 |
| 3-gram | 84.97 | 37.10 | 57.08 | 57.34 | 60.49 |
| 4-gram | 84.49 | 36.37 | 55.96 | 55.77 | 57.25 |
| 5-gram | 84.12 | 36.19 | 54.89 | 54.84 | 55.97 |

**Table 7:** The performance using different $n$-gram models in the SelfCheckGPT with $n$-gram method.

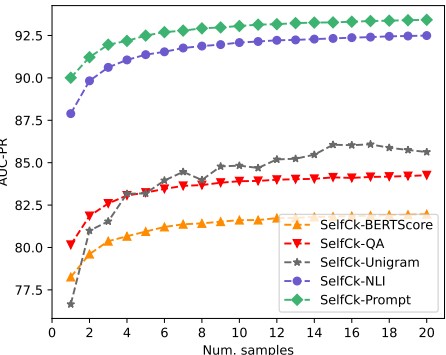

**Figure 8:** The performance of SelfCheckGPT methods on sentence-level non-factual detection (AUC-PR) versus the number of samples. This Figure extends the passage-level results in Figure 7.

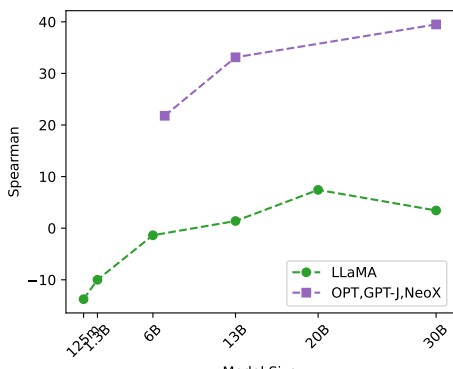

**Figure 9:** Passage-level ranking performance of the Avg($\mathcal{H}$) method using proxy LLM where the sizes are: LLaMA={7B, 13B, 30B}, OPT={125m, 1.3B, 13B, 30B}, GPT-J=6B, NeoX=20B. The full results are provided in Table 8.

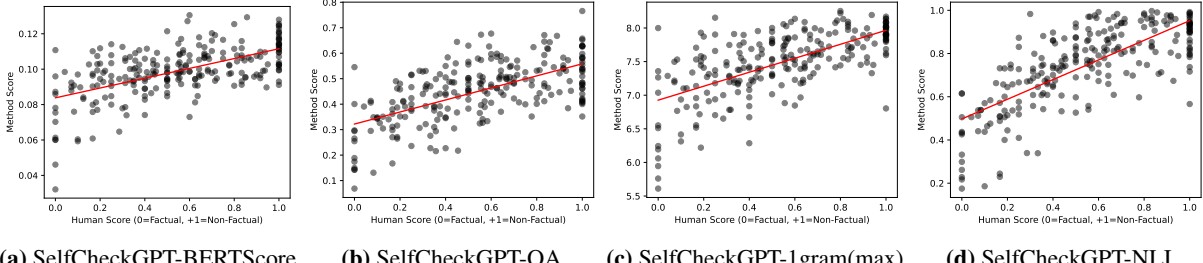

**(a)** SelfCheckGPT-BERTScore    **(b)** SelfCheckGPT-QA    **(c)** SelfCheckGPT-1gram(max)    **(d)** SelfCheckGPT-NLI

**Figure 10:** Scatter plot of passage-level scores where Y-axis = Method scores, X-axis = Human scores. Correlations are reported in Table 2. This figure provides results in addition to Figure 6.

| LLM | Size | Sentence-level (AUC-PR) | | | Passage-level (Corr.) | |
| | | NonFact | NonFact* | Factual | Pearson | Spearman |
|---|---|---|---|---|---|---|
| Random | - | 72.96 | 29.72 | 27.04 | - | - |
| *Avg(−log p) Method* | | | | | | |
| LLaMA | 30B | 75.43 | 30.32 | 41.29 | 21.72 | 20.20 |
| LLaMA | 13B | 74.16 | 30.01 | 37.36 | 13.33 | 12.89 |
| LLaMA | 7B | 71.69 | 27.87 | 31.30 | -2.71 | -2.59 |
| OPT | 30B | 67.70 | 24.43 | 25.04 | -32.07 | -31.45 |
| NeoX | 20B | 69.00 | 24.38 | 26.18 | -31.79 | -34.15 |
| OPT | 13B | 67.46 | 24.39 | 25.20 | -33.05 | -32.79 |
| GPT-J | 6B | 67.51 | 24.28 | 24.26 | -38.80 | -40.05 |
| OPT | 1.3B | 66.19 | 24.47 | 23.47 | -35.20 | -38.95 |
| OPT | 125m | 66.63 | 25.31 | 23.07 | -30.38 | -37.54 |
| *Avg(ℋ) Method* | | | | | | |
| LLaMA | 30B | 80.80 | 39.01 | 42.97 | 33.80 | 39.49 |
| LLaMA | 13B | 80.63 | 38.98 | 40.59 | 29.43 | 33.12 |
| LLaMA | 7B | 78.67 | 37.22 | 33.81 | 19.44 | 21.79 |
| OPT | 30B | 77.13 | 33.67 | 29.55 | -0.43 | 3.43 |
| NeoX | 20B | 77.40 | 32.78 | 30.13 | 5.41 | 7.43 |
| OPT | 13B | 76.93 | 33.71 | 29.68 | 0.25 | 1.39 |
| GPT-J | 6B | 76.15 | 33.29 | 28.30 | -2.50 | -1.37 |
| OPT | 1.3B | 74.05 | 31.91 | 26.33 | -10.59 | -10.00 |
| OPT | 125m | 71.51 | 30.88 | 25.36 | -14.16 | -13.76 |
| *Max(−log p) Method* | | | | | | |
| LLaMA | 30B | 74.01 | 27.14 | 31.08 | -22.83 | -22.71 |
| LLaMA | 13B | 71.12 | 26.78 | 28.82 | -34.93 | -31.70 |
| LLaMA | 7B | 69.57 | 25.91 | 26.54 | -42.57 | -38.24 |
| OPT | 30B | 67.32 | 24.40 | 24.32 | -49.51 | -45.50 |
| NeoX | 20B | 67.51 | 23.88 | 24.82 | -47.96 | -44.54 |
| OPT | 13B | 67.36 | 24.67 | 24.46 | -50.15 | -44.42 |
| GPT-J | 6B | 67.58 | 23.94 | 23.93 | -51.23 | -47.68 |
| OPT | 1.3B | 68.16 | 25.85 | 24.66 | -45.60 | -42.39 |
| OPT | 125m | 69.23 | 27.66 | 24.14 | -39.22 | -37.18 |
| *Max(ℋ) Method* | | | | | | |
| LLaMA | 30B | 80.92 | 37.32 | 37.90 | 35.57 | 38.94 |
| LLaMA | 13B | 80.98 | 37.94 | 36.01 | 32.07 | 34.01 |
| LLaMA | 7B | 79.65 | 35.57 | 31.32 | 22.10 | 22.53 |
| OPT | 30B | 76.58 | 33.44 | 29.31 | 1.63 | 6.41 |
| NeoX | 20B | 76.98 | 31.96 | 29.13 | 5.97 | 9.31 |
| OPT | 13B | 76.26 | 32.81 | 29.25 | 1.42 | 2.82 |
| GPT-J | 6B | 75.30 | 32.51 | 28.13 | -2.14 | 1.41 |
| OPT | 1.3B | 73.79 | 31.42 | 26.38 | -9.84 | -9.80 |
| OPT | 125m | 71.32 | 31.65 | 25.36 | -18.05 | -17.37 |

**Table 8:** AUC-PR for Detecting Non-Factual and Factual Sentences in the GPT-3 generated WikiBio passages. Passage-level PCC and SCC with LLMs used to assess GPT-3 responses. This table is an extension to Table 2.