# OpenReview forum: "SelfCheckGPT: Zero-Resource Black-Box Hallucination Detection for Generative Large Language Models"
_EMNLP/2023/Conference — EMNLP 2023 Main_

### Official Review · Reviewer_XVG4 · 2023-08-01

**Soundness:** 4

**Excitement:**

4: Strong: This paper deepens the understanding of some phenomenon or lowers the barriers to an existing research direction.

**Missing References:**

A self-consistency driven prompting method for LLM that is relevant to your method.

Xuezhi Wang, Jason Wei, Dale Schuurmans, Quoc Le, Ed Chi, and Denny Zhou. 2022. Self-consistency improves chain of thought reasoning in language models. arXiv preprint arXiv:2203.11171.

**Paper Topic And Main Contributions:**

This paper introduces a zero-resource hallucination detection method based on self-consistency of LLMs. They propose to evaluate stochastically-generated responses by LLMs and compare them with four similarity evaluation metrics. As no propoer evaluation sets exist, another major contribution of this paper is the presense of a sentence-level annotated hallucination detection dataset, consisting of 1908 sentences. Extensive experiments show the effectiveness of the proposed model.

**Reasons To Accept:**

- Point out an important yet under-evaluated task, hallucination detection.
- The methodology the authors propose seems intuitive, but the contribution of the dataset and associated experiments are more benefitial to the community.
- Experiments are solid, covering all the basis including the role of external knowledge.

**Reasons To Reject:**

- One potential reasons to reject is the limited scale and coverage of the dataset. But I believe it's a good inspiration to the community as a pilot work.

**Reproducibility:**

4: Could mostly reproduce the results, but there may be some variation because of sample variance or minor variations in their interpretation of the protocol or method.

**Reviewer Confidence:**

4: Quite sure. I tried to check the important points carefully. It's unlikely, though conceivable, that I missed something that should affect my ratings.

---

> ### Author Rebuttal · Authors · 2023-08-25
>
> Thank you Reviewer XVG4 for your useful feedback and suggestions, and thank you for the relevant reference, which we will incorporate into the revised version of this paper.

---

### Official Review · Reviewer_JhFY · 2023-08-08

**Soundness:** 3

**Excitement:**

4: Strong: This paper deepens the understanding of some phenomenon or lowers the barriers to an existing research direction.

**Paper Topic And Main Contributions:**

This paper investigates using LLMs to self-check their outputs for non-factual statements without resorting to fact-checking against reference data nor having access to output probability distributions. The authors propose 4 different methods (BERT score, QA, n-gram, prompting) of using an LLM to check the consistency of the generated output at the sentence level, which are then aggregated. The WikiBio dataset was annotated in this work at the sentence level into three factuality classes for evaluation, and experiments with GPT-3 show improvements over baselines, including grey-box approaches, with analysis on using different LLMs and comparison against fact-checking with external knowledge. The main contributions of this paper are:
1. Proposed using LLMs to self-check for factuality, which is black-box and doesn't use external resource.
2. Information consistency is checked on the sentence level, with 4 variants of measuring consistency proposed.
3. Curated a dataset based on WikiBio for evaluating hallucination.
4. Showed on GPT-3 generated text that the proposed method can outperform baselines on the curated dataset.

**Questions For The Authors:**

1. What are the other sampling parameters for the stochastically generated samples? For instance, say if the generated samples focus too much on diversity, it might become too inconsistent and adversely affect performance. Have you tried different sampling parameters?
2. Why take average for aggregation in section 7.2 instead of min/max? If a sentence in a passage is non-factual, it should indicate the entire passage contains non-factual info and thus the passage should be non-factual.

**Reasons To Accept:**

Checking LLM hallucinations is a very important topic, this paper proposes to use LLM to self-check (or external LLMs to assist). The authors also curated a fact-checking dataset that is used to show the effectiveness of the proposed method.

**Reasons To Reject:**

Although the curated dataset may be an asset to the community specifically for LLM hallucination/factuality, the evaluation is slightly weak as it was the only evaluated dataset, when multiple fact-checking datasets have commonly been used by the automatic fact-checking community (see Guo et al., 2022).

Guo et al. 2022: A Survey on Automated Fact-Checking, TACL

**Reproducibility:**

5: Could easily reproduce the results.

**Reviewer Confidence:**

4: Quite sure. I tried to check the important points carefully. It's unlikely, though conceivable, that I missed something that should affect my ratings.

---

> ### Author Rebuttal · Authors · 2023-08-25
>
> Thank you Reviewer JhFY for your useful feedback, queries and suggestions. We hope that our responses below are helpful in clarifying your questions and points:
>
> **Question 1**: A sampling parameter that may influence the responses is the generation temperature. In this work, we use the standard value and set the temperature to 1.0. If a temperature of 0.0 is used, then there is no diversity and self-check would not work. Similarly, for a high temperature, the generated samples would be highly diverse and even somewhat random, which can be unsuitable for generating evidence for self-check. With this insight, we hypothesize that there is a range of temperature values for which the selfcheck method can operate effectively. The paper selected the default value of temperature of 1.0 which provided effective performance, however, an interesting future ablation would be to plot detection performance against temperature.
>
>
> **Question 2**: We believe that taking the mean for aggregation is suitable because we want to capture the granularity of how factual a document/passage may be. Using a min or max operation for the document-level labels would effectively set binary labels of either non-factual or factual, which would lose fine-grained information about how factual the document is on a more fine-grained scale, which the mean operation provides.
>
> **Reason(s) to reject**: We agree that there are similarities between LLM hallucination detection and fact-checking, however, LLM hallucination aims to identify if an LLM has hallucinated, while fact verification aims to identify if a fact is true with respect to world knowledge. For fact-checking, the texts have already been created by humans or other automatic systems and are *not* generated from the LLM. This paper focuses specifically on LLM hallucination detection, and though it may be interesting to examine how the methods operate on fact verification tasks, that is not the aim of the paper. As a result, we had to curate a new evaluation dataset specifically for assessing LLM hallucination, and to the best of our knowledge, no other dataset for *real* LLM hallucination detection existed prior to our experiments.

---

### Official Review · Reviewer_91B1 · 2023-08-10

**Typos Grammar Style And Presentation Improvements:** None detected.
**Soundness:** 3

**Excitement:**

3: Ambivalent: It has merits (e.g., it reports state-of-the-art results, the idea is nice), but there are key weaknesses (e.g., it describes incremental work), and it can significantly benefit from another round of revision. However, I won't object to accepting it if my co-reviewers champion it.

**Missing References:**

No missing reference is noticed, the paper is well-situated in its context.

**Paper Topic And Main Contributions:**

The paper introduces a method to verify the factuality of sentences and paragraphs generated by LLMs, based on other utterances sampled from the same LLM for the same queries. The paper experimented with a few metrics for inconsistency, including BERT, a QA model, n-gram frequencies, and prompting an LLM. On a dataset they created based on WikiBio, their method demonstrated promising results compared with other "black box" or "grey box" approaches.

**Questions For The Authors:**

Question A: as the example in Figure 1 shows, pronouns are quite commonly involved in these descriptive paragraphs, how is coreference resolution handled in your approach, regarding the sentences taken from LLM generation, potentially containing isolated pronouns, and the sampled reference text?



**Reasons To Accept:**

The paper is well-written, related work is well-discussed, and experiments and ablation studies are thorough.

**Reasons To Reject:**

The weaknesses in this paper is primarily three-fold:

1) the proposed approach involves at least the re-generation of N samples with an LLM for each candidate generation to be verified, this comes at a high computational cost; for the prompt-based inconsistency measure, a further N*I number of queries are necessary;

2) the paper assumes the unavailability of the hidden states and output log probs when attempting to verify the factuality of LLM outputs; however, the likelihood of this assumed condition is not well-justified: for LLM service providers who have the responsibility to verify the generations, they would have access to the inner parameters; for the research community, it is ultimately more sound to make use of open-source LLMs such as LLaMA-2 / Falcon etc. in favor of such commercial products as ChatGPT;

3) with the black-box assumption, the authors excluded what they called "white-box solutions" from the comparison, such as (Azaria and Mitchell, 2023); however, it looks as though the white-box solutions are actually possible for the assumed setup, a similar way as the Proxy LLMs baseline. In fact, this would also be similar to the way the (Azaria and Mitchell) reference elicited their dataset in some domains.

Another weakness is, the paper claims to be the "first work to analyze model hallucination of general LLM responses", however their experiments are conducted on a rather restricted domain of WikiBio, with mostly factoid generations.


**Reproducibility:**

4: Could mostly reproduce the results, but there may be some variation because of sample variance or minor variations in their interpretation of the protocol or method.

**Reviewer Confidence:**

2: Willing to defend my evaluation, but it is fairly likely that I missed some details, didn't understand some central points, or can't be sure about the novelty of the work.

---

> ### Author Rebuttal · Authors · 2023-08-25
>
> Thank you Reviewer 91B1 for your useful feedback, queries and suggestions. We hope that our responses below are helpful in clarifying your questions and points:
>
> **Question A**: We do not explicitly handle coreference resolution, but since generated articles typically have one main topic and entire stochastic passages are used in selfcheck, resolving pronouns is likely to be simple for the approach to implicitly do (as observed with our strong results). However, coreference resolution is an interesting suggestion and would be a useful tool to incorporate in future applications of the method, particularly when responses have multiple persons/objects.
>
> **Reason To Reject 1**: We agree that Selfcheck-Prompt can be computationally heavy (as noted in the limitations). However, we believe the main contribution of our work is demonstrating an effective and simple solution for black-box zero-resource LLM hallucination, which has been underexplored and with no established solutions. In this paper, we have not focused on computational efficiency, though we note that there are simple ways to improve efficiency, for example through distillation. Alternatively, following experiments that use a small deberta-v3 classifier (with 435M params) trained on MNLI to predict entailment/contradiction yield competitive performance (shown in the table below), and require only $N$ API calls. Further, one could condition the stochastic samples on the main response, which could reduce the number of required samples. Nonetheless, we believe that the main contribution of the paper is its novel idea and demonstrated experimental results, which we believe may guide future research in LLM hallucination, where the method can be refined to be more effective as well as efficient.
>
> | Method                   |  NonFact (AUC-PR)  |  Factual (AUC-PR)  |   Ranking (PCC)   |
> |-----------------------------  |:------------------:|:------------------:|:-----------------:|
> | SelfCheck-NLI             |        92.50       |        66.08       |       74.14       |
> | SelfCheck-Prompt       |        93.42       |        67.09       |       78.32       |
>
> *We addressed a similar comment from Reviewer 3S1N which this response was based on.
>
> **Reason To Reject 2**: We would argue that LLMs are widely used by developers (who are not the LLM providers) who do not have access to the hidden states of LLMs like GPT3. Therefore, integrating hallucination detection into their applications requires a black-box approach. Secondly, we selected GPT3 for our investigation as it is widely used in both industry and academia. This system is considered to be much more capable at text generation than existing open-source LLMs, which was especially true when our experiments were conducted (when Llama2 and Falcon did not exist). We note that although our methods work for black-box systems, black-box solutions are the most general and are applicable to grey/white-box systems, and our results demonstrate that it can outperform grey-box baselines.
>
> **Reason To Reject 3**: Our results show that there is poor alignment between the proxy LLM and the system probabilities, which implies that it is unlikely that the proxy hidden representations would align well. Furthermore, the methods in Azaria and Mitchell, 2023 are white-box and supervised approaches, which require training data to train a classifier using the hidden representations, limiting the robustness and generalizability of the method.
>
> **Reason To Reject 4**: What we meant by “the first work to analyze model hallucination of general LLM responses” is that although previous works have looked at identifying hallucination in NLG tasks (e.g. summarization), none consider hallucination detection of LLM responses when LLMs are probed to freely generate responses to probed queries. We agree that our wording here can be confusing, and we will make changes in future versions to make this statement clearer.

---

### Official Review · Reviewer_3S1N · 2023-08-11

**Soundness:** 4

**Excitement:**

4: Strong: This paper deepens the understanding of some phenomenon or lowers the barriers to an existing research direction.

**Paper Topic And Main Contributions:**

This paper proposed SelfCheckGPT, a black-box, zero-resource method for LLM hallucination detection. This approach utilizes multiple sampled responses to verify the information consistency. The primary contributions include:
1. A zero-resource method for detecting hallucinations in LLM responses, alongside four SelfCheckGPT variations: BERTScore, question-answering, n-gram, and LLM prompting.
2. The release of an annotated dataset for sentence-level hallucination detection in LLM.
3. The proposed method achieves notable results in comparison to baseline approaches, all without requiring supplementary knowledge bases or token-level probabilities.

**Questions For The Authors:**

1. I couldn't locate any explanation regarding the random baseline. After reviewing the code, I noticed that it's simply based on calculating the proportion of positive samples, rather than a random 0-1 label assigned to each sentence. I wonder is it appropriate to call it "random"?
2. For the non-factual* setting, the minor-inaccurate samples are all categorized as negative ones (label 0), right?

**Reasons To Accept:**

1. This paper provides a strong baseline for the emerging field of LLM hallucination detection, offering insights for future related research.
2. The proposed method exclusively relies on LLM responses, eliminating the necessity for supplementary knowledge or access to LLM's internal states.
3. Each of the four variants of SelfCheckGPT demonstrates impressive performance gains in contrast to the baselines, across both sentence-level and passage-level evaluations.
4. The author released WikiBio GPT-3 dataset comprising 238 passages and 1908 annotated sentences. This resource stands to facilitate researchers in delving into the intricacies of LLM hallucination.
5. The experiment section is meticulously designed, providing strong support for the author's insights.
6. This paper is well written.

**Reasons To Reject:**

1. The method presented relies on extracting multiple responses from the LLM. For the variant with optimal performance, LLM prompting, 20 samples are needed to achieve the best reported results. Assuming a response contains 5 sentences, this requires 100 API calls to obtain a passage-level score (if I understand correctly), which is cost heavy and ineffective.
2. It remains unclear whether the proposed approach is suitable for detecting hallucinations in responses from other LLMs and across various application scenarios beyond WikiBio. This uncertainty arises because the experiment dataset exclusively encompasses WikiBio responses drawn from text-davinci-003.
3. The proposed method might struggle to detect hallucinations in open-ended responses, for example, the prompt "introduce a sports celebrity to me". In this case, the sampled responses could pertain to different individuals, making it challenging to identify shared information for consistency checking.

**Reproducibility:**

5: Could easily reproduce the results.

**Reviewer Confidence:**

5: Positive that my evaluation is correct. I read the paper very carefully and I am very familiar with related work.

---

> ### Author Rebuttal · Authors · 2023-08-25
>
> Thank you Reviewer 3S1N for your helpful feedback and questions. We hope that our responses below address/answer your comments:
>
> **Question 1**: What we mean by “random baseline” is that the baseline randomly predicts the positive class (e.g., non-factual) with probability $p$ (and negative class with prob. $1-p$) on any sentence. By doing this, the expected precision = the proportion of the positive examples, while recall = $p$. Thus, when sweeping $p$ from 0.0 to 1.0, it gives a horizontal line in Figure 5 and therefore AUC-PR = the proportion of the positive examples. More explanation of the random baseline can be found in this StackOverflow article: https://stats.stackexchange.com/a/266989.
>
> **Question 2**: Yes, in the non-factual* setup, minor-inaccurate samples are assigned the negative label (0).
>
> **Reason To Reject 1**: We agree that Selfcheck-Prompt can be computationally heavy (as noted in the limitations). However, we believe the main contribution of our work is demonstrating an effective and simple solution for black-box zero-resource LLM hallucination, which has been underexplored and with no established solutions. In this paper, we have not focused on computational efficiency, though we note that there are simple ways to improve efficiency, for example through distillation. Alternatively, following experiments that use a small deberta-v3 classifier (with 435M params) trained on MNLI to predict entailment/contradiction yield competitive performance (shown in the table below), and require only $N$ API calls for obtaining samples. Further, one could condition the stochastic samples on the main response, which could reduce the number of required samples. Nonetheless, we believe that the main contribution of the paper is its novel idea and demonstrated experimental results, which we believe may guide future research in LLM hallucination, where the method can be refined to be more effective as well as efficient.
>
> | Method                   |  NonFact (AUC-PR)  |  Factual (AUC-PR)  |   Ranking (PCC)   |
> |-----------------------------  |:------------------:|:------------------:|:-----------------:|
> | SelfCheck-NLI             |        92.50       |        66.08       |       74.14       |
> | SelfCheck-Prompt       |        93.42       |        67.09       |       78.32       |
>
>
> **Reasons to Reject 3**: We agree that our method was shown to be effective for the particular task setup. Although the current version of the work may not be applicable to all scenarios, as mentioned in our previous response, there are simple extensions that could be applied to increase the generalisability and effectiveness of the approach. For example, one could generate samples conditioned on the original response so that one can align stochastic samples with information to be checked, e.g., if for the prompt `introduce a sports celebrity to me`, the system generates the response`Lionel Messi is …`, one can then dynamically prompt the LLM to generate facts/information about `Lionel Messi` and apply selfcheck in the provided scenario (in an automated fashion).

---

### Meta-Review · Area_Chair_YqZE · 2023-09-16

**Recommendation:** 5

**Metareview:**

This paper investigates hallucination detection from LLMs, introducing a formal version of the task and creating strong baselines. This will encourage future work both through methods and through the provided dataset. The only noteworthy negatives identified in the review process were some assumptions of what is required from the underlying LLMs. However, these assumptions do not seem impossible for most models.

---

### Decision · Program_Chairs · 2023-10-07

**Decision:**

Accept-Main

**Comment:**

This paper investigates hallucination detection from LLMs, introducing a formal version of the task and creating strong baselines. This will encourage future work both through methods and through the provided dataset. The only noteworthy negatives identified in the review process were some assumptions of what is required from the underlying LLMs. However, these assumptions do not seem impossible for most models.